# Harmful and Harmless Soil-Dwelling Fungi Indicate Microhabitat Suitability for Off-Host Ixodid Ticks

**DOI:** 10.3390/microorganisms12030609

**Published:** 2024-03-19

**Authors:** Claire E. Gooding, Layla Gould, Gerhard Gries

**Affiliations:** Department of Biological Sciences, Simon Fraser University, Burnaby, BC V5A 1S6, Canada; cgooding@sfu.ca (C.E.G.);

**Keywords:** *Beauveria bassiana*, entomopathogenic fungus, hard ticks, off-host microhabitat, 2-methylisoborneol

## Abstract

Following blood meals or questing bouts, hard ticks (Ixodidae) must locate moist off-host microhabitats as refuge. Soil-dwelling fungi, including entomopathogenic *Beauveria bassiana* (*Bb*), thrive in moist microhabitats. Working with six species of ixodid ticks in olfactometer bioassays, we tested the hypothesis that ticks avoid *Bb*. Contrary to our prediction, nearly all ticks sought, rather than avoided, *Bb*-inoculated substrates. In further bioassays with female black-legged ticks, *Ixodes scapularis*, ticks oriented towards both harmful *Bb* and harmless soil-dwelling fungi, implying that fungi—regardless of their pathogenicity—signal habitat suitability to ticks. Only accessible *Bb*-inoculated substrate appealed to ticks, indicating that they sense *Bb* or its metabolites by contact chemoreception. *Bb*-inoculated substrate required ≥24 h of incubation before it appealed to ticks, suggesting that they respond to *Bb* metabolites rather than to *Bb* itself. Similarly, ticks responded to *Bb*-inoculated and incubated cellulose but not to sterile cellulose, indicating that *Bb* detection by ticks hinges on the *Bb* metabolism of cellulose. 2-Methylisoborneol—a common fungal metabolite with elevated presence in disturbed soils—strongly deterred ticks. Off-host ticks that avoid disturbed soil may lower their risk of physical injury. Synthetic 2-methylisoborneol could become a commercial tick repellent, provided its repellency extends to ticks in diverse taxa.

## 1. Introduction

Ticks (Ixodida) are ectoparasites of vertebrates with diverse life histories, host-seeking strategies, and habitat preferences [1]. Most soft ticks (Argasidae) are nidicolous and remain within nests or burrows of a host throughout their entire life, repeatedly feeding on the same host species [2]. Hard ticks (Ixodidae) are typically non-nidicolous, seek hosts outside burrows, and typically engage in a ‘questing’ ambush strategy, climbing onto hosts as they pass by [3,4,5,6,7]. Alternatively, some species actively pursue hosts [8]. Unlike nidicolous ticks, non-nidicolous ticks must locate suitable off-host microhabitats to rest following questing attempts or blood meals. The selection of these microhabitats is affected by tick-intrinsic physiological factors, arrestment pheromones, and microhabitat characteristics [9,10].

The behavior, survival, and distribution of ixodid ticks are all affected by the availability and quality of questing locations and off-host microhabitats. The timing and duration of questing are dependent upon ambient relative humidity and sunlight exposure [11,12,13]. While questing, ticks lose water and periodically must replenish it in humid leaf litter and detritus [11]. High humidity, low sunlight exposure, and sufficient organic matter (e.g., leaf litter) to provide refuge are key requisites of suitable off-host microhabitats for ticks [14,15,16,17,18,19].

The microbiome of resource sites informs foraging decisions of arthropods both directly and indirectly. For example, nectar-dwelling microbes affect nectar-foraging decisions of mosquitoes (direct effect) [20,21], whereas the root microbiome of plants affects aboveground herbivory (indirect effect) [22,23]. Moreover, many dipterans, including mosquitoes and stable flies, select oviposition sites based on their microbial community [24,25,26,27]. The attraction of *Ixodes* ticks to their vertebrate hosts is mediated, in part, by volatiles emitted from host skin microbiota [28]. However, whether the microbiota in the soil and leaf litter informs the selection of off-host microhabitats by ticks is largely unknown. Other soil-dwelling arthropods select microhabitats based on the presence of certain microorganisms. For example, springtails, *Folsomia candida*, are attracted to soil colonized by edible bacteria [29], whereas red imported fire ant queens, *Solenopsis invicta,* preferentially nest in soil colonized by bacteria that inhibit the growth of entomopathogenic fungi [30]. Both springtails and queen red imported fire ants are attracted to the sesquiterpenoid geosmin and the monoterpene 2-methylisoborneol (2-MIB) emitted by cyanobacteria [25,31,32], actinobacteria [29,30,33], and fungi [34,35,36].

Moist off-host microhabitats offer not only abiotic benefits to ticks but also present biotic threats from pathogens and predators. To reduce predation risk, many arthropods, including *Ixodes scapularis* ticks [37], exploit chemical cues indicative of predator presence [37,38,39,40,41,42,43,44]. Entomopathogenic fungi, such as *Beauveria bassiana* (Bals.-Criv.) Vuill. (1912) (*Bb*) dwelling in soil and detritus [45,46,47], are lethal to ticks and other arthropods [48,49,50]. To reduce the risk of fungal infections, some arthropods avoid sites colonized by harmful fungi [51,52,53,54,55,56] but occasionally may even be attracted to them [57,58]. Arthropods may detect harmful fungi by sensing their volatile metabolites (olfaction) or by recognizing specific chemicals on the fungal surface (contact chemoreception). Ticks sense semiochemicals (message-bearing chemicals) using sensory receptors on their front legs and/or on their palps [59]. Conversely, harmless fungi present in shaded, damp litter and detritus [60,61,62,63] may be valuable indicators of suitable moist microhabitats that off-host ticks seek for refuge. The behavioral responses of ticks to harmless and harmful (entomopathogenic) fungi have not yet been investigated.

Here, we worked with females and males of six species of ixodid ticks that are taxonomically diverse and of medical and/or veterinary importance: the lone star tick, *Amblyomma americanum*, American dog tick, *Dermacentor variabilis*, brown dog tick, *Rhipicephalus sanguineus*, castor bean tick, *Ixodes ricinus*, western black-legged tick, *Ixodes pacificus*, and black-legged tick, *Ixodes scapularis.* We tested the hypothesis that these ticks avoid substrate inoculated with *Bb*. With our data revealing that ticks seek, rather than avoid, *Bb*-inoculated substrate, we then studied underlying mechanisms for their behavioral responses, working with female *I. scapularis* as representative model organisms. Specifically, we tested whether preferential responses by female *I. scapularis* to *Bb*-inoculated substrate are dependent upon (*i*) the *Bb* incubation period, (*ii*) olfactory or contact-chemoreceptive recognition of *Bb* or its metabolites, and (*iii*) the presence of cellulose as a *Bb* growth medium. We further investigated whether female *I. scapularis* also seeks harmless soil-dwelling fungi as indicators of suitable off-host refuges and whether common volatiles of soil-dwelling fungi attract ticks to off-host refuges.

## 2. Methods

### 2.1. Tick Maintenance

Adult males and females of *I. scapularis*, *I. ricinus*, *I. pacificus*, *D. variabilis*, *A. americanum*, and *R. sanguineus* were obtained from BEI Resources (American Type Culture Collection), and additional *I. scapularis* adults were purchased from the National Tick Research and Education Resource (Oklahoma State University). We tested adult ticks because—based on our experience—they tolerate laboratory conditions better than immature ticks. Ticks were maintained at 22 °C uer a 14:10 light/dark cycle and housed singly in 1.5-mL microcentrifuge tubes (Corning Inc., Reynosa, MX, USA) with a mesh-covered 5-mm hole in the lid for ventilation. Sets of 10–20 tubes were held in 150-mL plastic cups with lids that, in turn, were enclosed in clear plastic bins (46 × 32 × 18 cm). Damp cotton rounds (Dollarama, QC, Canada) placed into cups provided sufficient humidity (80–90% RH). Every week, cotton rounds were replaced, and any deceased ticks were removed to prevent potential infections.

### 2.2. Propagation of Fungi and Collection of Conidia

*Beauveria bassiana* (GHA) was provided by the Cory lab at Simon Fraser University, and *Fusarium oxysporum*, *Penicilium roqueforti*, and *Rhizopus stolonifer* were purchased from Merlan Scientific (Toronto, ON, Canada). The fungi were propagated on Sabouraud dextrose agar (SDA) plates (90 × 15 mm), which were sealed with parafilm (Bemis, Sheboygan Falls, WI, USA) and incubated 7 days at 26–28 °C. To harvest conidia, each plate was flooded with 50 mL of sterile 0.05% Tween80 (Sigma Aldrich, St. Louis, MO, USA) and gently scrubbed using a sterile inoculating loop to create a conidial suspension. This suspension was then filtered through sterile glass wool to minimize mycelial or agar debris, and the conidial suspension was vortexed for 30 s to homogenize it. The concentrations of the conidial suspensions were determined using a Neubauer improved Hemocytometer (Superior Marienfeld, Lauda-Königshofen, Germany) and diluted to a concentration of 10^7^ conidia/mL for all experiments. The dose of 10^7^ conidia/mL elicited the strongest behavioral responses in a preliminary dose–response experiment (see Appendix A).

### 2.3. Preparation of Experimental Substrates

Sterile dry coconut fiber (PetSmart, Phoenix, AZ, USA) was placed in 5-mL aliquot doses and in 50-mL sterile centrifuge tubes (Cornell, Tamaulipas, MX, USA), which were inoculated with either a 1-mL conidial suspension (treatment) or a 1-mL sterile 0.05% Tween80 solution (control). The centrifuge tubes were incubated for 24 h at 26–28 °C, unless otherwise noted. Coconut fiber was used as a fungal growing medium because it comprises constituents (cellulose, lignin, and hemicelluloses [64]) that are common in partially decomposed plant matter, where ticks are likely to take refuge [65].

To produce a substrate texturally similar to coconut fiber and based only on cellulose, filter paper discs (90 mm diam.) were shredded for 5 min in an Osterizer 10-speed blender (Sunbeam Products, Boca Raton, FL, USA), producing 2- to 5-mm pieces. Aliquots (5 mL) of shredded filter paper were placed in centrifuge tubes and inoculated with either a 1-mL conidial suspension (treatment) or a 1-mL sterile 0.05% Tween80 solution (control).

### 2.4. General Experimental Design

All behavioral responses of ticks to experimental substrates were tested in two-choice still-air olfactometers (150 × 50 × 17 mm; Figure 1). We used still-air, instead of moving-air, olfactometers because off-host ticks encounter fungi in soil microhabitats such as leaf litter, where there is typically little, if any, air movement. Each olfactometer had three inset circular chambers (ID = 28 mm) inter-connected with inset linear paths (24 × 10 × 7 mm). The central chamber had a depth of 7 mm, whereas the lateral chambers had a depth of 16 mm with a 2-mm wide lip of 9-mm depth to accommodate a 9-mm watch glass. The olfactometers were modeled and 3D-printed using Autodesk Fusion 360 (Version 13.2.0.9150), Creality Slicer (Version 4.8.2), and an Ender-3 Pro 3D printer (Creality, Shengzhen, China). The olfactometers were printed using translucent 1.75-mm (±0.03 mm dimensional accuracy) polylactic acid (PLA) filament (GIANTARM, Shengzhen, China). To reduce the porosity of 3D-printed olfactometers, we applied XTC-3D brush-on epoxy coating (Smooth-On Inc., Macungie, PA, USA). The stereolithography (STL) file used for 3D printing is included in Appendix A.

Treatment and control experimental substrates were assigned to lateral chambers, alternating the position of stimuli between replicates to account for potential side bias. Substrates were poured into the wells of lateral chambers and leveled to create a uniformly flat surface. To initiate an experimental replicate, a single tick was introduced into the central chamber, briefly exposed to human exhale to stimulate movement, and then allowed 30 min to respond. A 30-min bioassay time was deemed sufficient because, in pre-screening tests, 80% of female *I. scapularis* left the central olfactometer chamber within 30 min. The olfactometers were sealed with parafilm and a rectangular lid (150 × 50 × 3 mm). A tick was considered a responder if it was found in a lateral chamber after 30 min. All other ticks were deemed non-responders and excluded from statistical analyses but were reported in figures. After each experiment, the olfactometers were cleaned with 70% ethanol and hexane. At the end of each experimental day, the olfactometers were washed with Sparkleen (Thermo Fisher Scientific, Waltham, MA, USA), rinsed with distilled water, and air-dried.

### 2.5. Specific Experiments

#### 2.5.1. Effect of *Bb*-Inoculated Substrate on Behavioral Responses of Diverse Tick Taxa

To investigate whether harmful *Bb* is a generic deterrent to diverse taxa of ticks, experiments 1–12 tested the behavioral responses of adult female and male *A. americanum, D. variabilis, R. sanguineus, I. ricinus*, *I. pacificus*, and *I. scapularis* to coconut fiber inoculated or not inoculated (control) with a *Bb* conidial suspension. Thirty replicates were run for each sex of each species, but 40 replicates were run for female and male *R. sanguineus* due to a high rate of non-responding ticks.

#### 2.5.2. Effect of *Bb*-Incubation Period on Behavioral Responses of Ticks

After obtaining evidence that ticks—unexpectedly—sought, rather than avoided, *Bb*-inoculated coconut fiber (see Section 3), we investigated whether the ticks’ preferential responses to *Bb*-inoculated fiber were contingent upon the *Bb*-incubation period. To this end, experiments 13–17 (*n* = 30 each) tested the behavioral responses of female *I. scapularis* to fiber inoculated or not inoculated (control) with a conidial suspension of *Bb* incubated at 26–28 °C for 0, 24, 48, 72, and 96 h prior to bioassays.

#### 2.5.3. Olfactory or Contact-Chemoreceptive Recognition of *Bb* (or Its Metabolites) by Ticks

To investigate whether the preferential responses of ticks to *Bb*-inoculated coconut fiber (see Section 3) were mediated by olfaction (i.e., receptors sensing airborne semiochemicals) or by contact chemoreception (i.e., receptors sensing substrate-borne stimuli through physical contact), parallel experiments 18–19 (*n* = 30 each) tested the responses of ticks to coconut fiber, which was physically accessible (Exp. 18) or inaccessible (Exp. 19). Coconut fiber was made inaccessible by placing a mesh screen (28 mm diam.) in each lateral chamber and by sealing it around its edge with plasticine to prevent ticks from passing under the screens. Test stimuli in both experiments consisted of fiber inoculated or not inoculated (control) with *Bb*.

#### 2.5.4. Effect of Cellulose or Its Fungal Metabolites on Behavioral Responses of Ticks

Because ticks favorably responded to *Bb*-inoculated coconut fiber only after incubation/metabolism for at least 24 h (see Section 3), it was conceivable that ticks responded to *Bb* metabolites of cellulose, a constituent in experimental coconut fiber (cellulose, lignin, and hemicelluloses) and a major component of plant cell walls that fungi commonly metabolize. To investigate this concept, parallel experiments 20 and 21 tested the responses of female *I. scapularis* to the cellulose-only substrate (Exp. 20), and coconut fiber (Exp. 21), with either substrate in both experiments inoculated (treatment) or not inoculated (control) with *Bb*.

#### 2.5.5. Effect of Various Soil-Dwelling Fungi on Behavioral Responses of Ticks

To investigate whether not only *Bb* but also other soil-dwelling fungi elicit preferential responses by ticks, experiments 20–23 (*n* = 30 each) offered female *I. scapularis* a choice between coconut fiber inoculated or not inoculated with a 1-mL 10^7^ conidia/mL suspension of *R. stolonifer* (Exp. 22), *F. oxysporum* (Exp. 23), *P. roqueforti* (Exp. 24), and *B. bassiana* (Exp. 25).

#### 2.5.6. Effect of Fungus-Derived Volatiles on Attraction of Ticks

As ticks sought substrate inoculated with various species of soil-dwelling fungi (see Section 3) and drawing on reports that springtails and *S. invicta* queen and worker ants are attracted to 2-methylisoborneol (2-MIB) and geosmin, which are commonly emitted by fungi [34,35,36], we further investigated whether ticks are also attracted to 2-MIB and geosmin. To this end, we placed a watch glass (28 mm diam.) fitted with a congruent piece of filter paper in lateral olfactometer chambers and applied 2-MIB dissolved in 25 µL of methanol at doses of 1.0 ng (Exp. 26), 0.1 ng (Exp. 27), and 0.01 ng (Exp. 28) to the treatment filter paper, and 25 µL of methanol to the corresponding control filter paper. Similarly, we tested geosmin dissolved in 25 µL of methanol at doses of 1.0 ng (Exp. 29), 0.1 ng (Exp. 30), and 0.01 ng (Exp. 31), using 25 µL of methanol as the control stimulus. In each experimental replicate, methanol was allowed 5 min to evaporate before a tick was introduced into the central olfactometer chamber. The 2-MIB and geosmin dose range of 0.01–1 ng was deemed ecologically relevant because *S. invicta* worker and queen ants were attracted to geosmin at a dose of 2 ng [30], and analyses of above-soil headspace volatiles yielded 0.2–9.0 ng/mL of 2-MIB and 0.01–0.7 ng/mL of geosmin over 20–30 min [66]. Geosmin and 2-MIB were purchased from Sigma-Aldrich (St. Louis, MO, USA).

### 2.6. Statistical Analysis

All data were analyzed using RStudio (2023.03.1+446), and all figures were prepared using RStudio and Inkscape (Version 1.2.2). The scales package [67] was used to aid in creating figures. The ticks’ responses to bioassay stimuli were analyzed by comparing the ratio of treatment and control responses to a hypothetical response ratio of 1:1, using a two-sided exact binomial test and excluding non-responders from analyses.

## 3. Results

### 3.1. Effect of Bb-Inoculated Substrate on Behavioral Responses of Ticks

All ticks—except for female and male *R. sanguineus* and female *I. pacificus*—tested in experiments 1–12 (Figure 2) preferred *Bb*-inoculated coconut fiber to sterile (control) fiber [*A. americanum*; Exp. 1 (females): *n* = 27, *p* = 0.0059; Exp. 2 (males): *n* = 27, *p* < 0.0001; *D. variabilis*; Exp. 3 (females): *n* = 23, *p* = 0.035; Exp. 4 (males): *n* = 21, *p* = 0.027; *R. sanguineus*; Exp. 5 (females): *n* = 24, *p* = 0.54; Exp. 6 (males): *n* = 18, *p* = 0.096; *I. ricinus*; Exp. 7 (females): *n* = 28, *p* = 0.00091; Exp. 8 (males): *n* = 25, *p* = 0.00091; *I. pacificus*; Exp. 9 (females): *n* = 28, *p* = 0.18; Exp. 10 (males): *n* = 17, *p* = 0.013; *I. scapularis*; Exp. 11 (females): *n* = 24, *p* = 0.0026; Exp. 12 (males): *n* = 26, *p* = 0.0025]. Altogether, the data indicate a widespread failure of ticks in diverse taxa to avoid a harmful entomopathogenic fungus.

### 3.2. Effect of Bb-Incubation Period on Behavioral Responses of Ticks

Without prior incubation of *Bb*-inoculated coconut fiber, female *I. scapularis* responded equally to *Bb*-inoculated coconut fiber and sterile fiber (Exp. 13: *n* = 20, *p* = 0.15; Figure 3). In contrast, after an incubation period of 24, 48, 72, and 96 h, female *I. scapularis* invariably preferred *Bb*-inoculated coconut fiber to sterile fiber [24 h (Exp. 14): *n* = 26, *p* < 0.0001; 48 h (Exp. 15): *n* = 19, *p* < 0.0001; 72 h (Exp. 16): *n* = 25, *p* < 0.0001; 96 h (Exp. 17): *n* = 25, *p* = 0.0002; Figure 3]. The data suggest that the responses of ticks are likely mediated, in part, by fungal metabolites produced during incubation.

### 3.3. Olfactory or Contact-Chemoreceptive Recognition of Bb (or Its Metabolites) by Ticks

Physical access to coconut fiber inoculated (treatment) or not inoculated (control) with *Bb* determined the ticks’ behavioral responses (Figure 4). When access to coconut fiber was blocked, female *I. scapularis* responded equally to treatment and control fibers (Exp. 18: *n* = 24, *p* = 0.31). However, when access was not blocked, females preferred *Bb*-inoculated fiber to sterile fiber (Exp. 19: *n* = 25, *p* = 0.015), indicating that recognition of *Bb* is based on contact chemoreception rather than olfaction.

### 3.4. Effect of Cellulose or Its Fungal Metabolites on Behavioral Responses of Ticks

Cellulose as a *Bb* culture medium was sufficient to elicit preferential responses by ticks (Figure 5). Cellulose inoculated with *Bb* (Exp. 20), and coconut fiber (consisting of cellulose, lignin, and hemicelluloses) inoculated with *Bb* (Exp. 21), both elicited stronger behavioral responses from female *I. scapularis* than the corresponding sterile cellulose (Exp. 20: *n* = 25, *p* = 0.043) or sterile coconut fiber (Exp 25: *n* = 29, *p* = 0.024). These data, coupled with those presented in Figure 3, suggest that the *Bb* breakdown of cellulose contributes to the preferential foraging responses of ticks.

### 3.5. Effect of Various Soil-Dwelling Fungi on Behavioral Responses of Ticks

All four species of fungi tested elicited preferential responses by ticks (Figure 6). Relative to sterile coconut fiber, female *I. scapularis* preferred coconut fiber inoculated with *R. stolonifer* (Exp. 22: *n* = 26, *p* = 0.0094), *F. oxysporum* (Exp. 23: *n* = 23, *p* = 0.011), *P. roqueforti* (Exp. 24: *n* = 25, *p* = 0.0041), and *Bb* (Exp. 25: *n* = 19, *p* = 0.019).

### 3.6. Effect of Fungus-Derived Volatiles on Attraction of Ticks

2-Methylisoborneol (2-MIB), but not geosmin, affected the behavioral responses of ticks (Figure 7). Female *I. scapularis* were strongly deterred by 2-MIB at a dose of 1 ng (Exp. 26: *n* = 21, *p* = 0.0002), slightly deterred at a dose of 0.1 ng (Exp. 27: *n* = 24, *p* = 0.064), and undeterred at a dose of 0.01 ng (Exp. 28: *n* = 17, *p* = 1). Conversely, irrespective of the dose tested, female *I. scapularis* ticks were neither deterred by, nor attracted to, geosmin [1 ng (Exp. 29): *n* = 19, *p* = 0.65; 0.1 ng (Exp. 30): *n* = 23, *p* = 1; 0.01 ng (Exp. 31): *n* = 23, *p* = 0.21].

## 4. Discussion

Our data do not support the hypothesis that ixodid ticks (*A. americanum, D. variabilis, R. sanguineus, I. ricinus*, *I. pacificus*, and *I. scapularis*) avoid substrate inoculated with the harmful entomopathogen *Beauveria bassiana* (*Bb*). Contrary to our prediction, all ticks tested—except for female and male *R. sanguineus* and female *I. pacificus*—preferred *Bb*-inoculated coconut fiber to sterile fiber, implying that the benefits accruing from these preferential responses outweigh the potential harm inflicted on ticks by *Bb*. The fact that female and male *R. sanguineus,* which are known to inhabit human dwellings [68], were indifferent to the presence of *Bb* may be attributed to their reduced reliance on moist micro-habitats, as typically indicated by the presence of fungi.

Ticks orienting towards harmful *Bb* and towards harmless *R. stolonifer*, *F. oxysporum*, and *P. roqueforti* may accrue benefits in that all these fungi—regardless of their pathogenicity—might signal habitat suitability for ticks. High humidity, low sunlight exposure, and the presence of organic matter (e.g., leaf litter) for refuge are all essential requisites of favorable off-host microhabitats for ticks [14,15,16,17,18,19]. As fungal biomass in the soil is positively correlated with soil moisture [63] and negatively correlated with sunlight exposure [61,62], fungal presence or high relative fungal biomass could be a measure of sufficiently moist and shaded microhabitats that ticks require for their well-being and survival. Through both hygroreception and photoreception, ticks are able to gauge the current suitability of a microhabitat [69], but they cannot readily gauge sustained microhabitat suitability. However, fungal breakdown products of plant cellulose could reliably indicate long-term habitat suitability because the metabolism of plant cellulose by fungi [70] and the ensuing accumulation of high fungal biomass are supported by relatively persistent moisture and shade [71].

Preferential responses of ticks to *Bb*-inoculated fiber only when it was physically accessible indicate that ticks sense *Bb,* or its metabolites, mainly by contact chemoreception. Compounds sensed by contact chemoreceptors typically have high molecular weight and low volatility, and thus cannot readily be detected by olfaction [72]. It is plausible, however, that the ticks’ preferential responses to *Bb*-inoculated coconut fiber are mediated by both contact chemoreception and olfaction because—numerically—more ticks oriented to *Bb*-inoculated fiber than to sterile fiber, even when access to coconut fiber was blocked.

Without at least 24-h incubation of *Bb*-inoculated coconut fiber, ticks responded equally to *Bb*-inoculated fiber and to sterile fiber. These findings imply that ticks respond to products of *Bb* metabolism rather than to *Bb* itself. The fact that ticks responded equally to *Bb*-inoculated coconut fiber (consisting of cellulose, lignin, and hemicelluloses) and to *Bb*-inoculated cellulose further implies that it is the breakdown products of cellulose that mediate *Bb* recognition.

There is no obvious explanation as to why ticks avoided 2-MIB but responded indifferently to geosmin. Both 2-MIB and geosmin are emitted by numerous microorganisms, including cyanobacteria [25,30,31], actinobacteria [29,30,33], and fungi [34,35,36], and both compounds commonly co-occur in above-soil headspace [66]. Based on current literature, there are no consistent behavioral responses of arthropods to 2-MIB and geosmin. Geosmin attracts the yellow fever mosquito *Aedes aegypti* to oviposition sites [25,29,30]. Sporulating streptomyces bacteria emit geosmin and 2-MIB, thereby attracting springtails that then aid spore dispersal [25,29,30]. Newly mated queens of *S. invicta* are attracted to geosmin and 2-MIB produced by actinobacteria as indicators of suitable soil nesting sites, with reduced risk of entomopathogenic fungal infections [30]. Conversely, *Drosophila melanogaster* vinegar flies sense and avoid geosmin as an indicator of feeding or oviposition sites containing harmful bacteria [73], and the bacteriophagous nematode *Caenorhabditis elegans* avoids grazing on geosmin-producing bacteria [74]. As 2-MIB was a strong deterrent to ticks only at elevated levels, microhabitats with elevated 2-MIB levels may signal potential threats that ought to be avoided. Ticks encounter elevated 2-MIB levels in disturbed soils, which produce relatively large amounts of 2-MIB and geosmin [66,75]. As any form of current or future soil disturbance may physically harm off-host ticks, it follows that the avoidance of sites with a strong 2-MIB odor is adaptive to ticks.

In conclusion, the cues that moisture-dependent hard ticks exploit to locate and select suitable off-host microhabitats were previously not known. We present data showing that the presence of soil-dwelling fungi (or their metabolites)—irrespective of their pathogenicity—informs decisions of ticks that seek suitable off-host microhabitats. Avoiding sites with elevated 2-MIB levels may help off-host ticks reject sites prone to significant disturbance that is harmful to ticks. The fact that 2-MIB is a deterrent to ticks was an unexpected and serendipitous finding in our study. Synthetic 2-MIB alone, or in combination with other tick deterrents [37,76,77], may become a highly effective commercial tick repellent, provided that the repellent effect extends to ticks in diverse taxa. This line of research is currently ongoing.

## Figures and Tables

**Figure 1 microorganisms-12-00609-f001:**
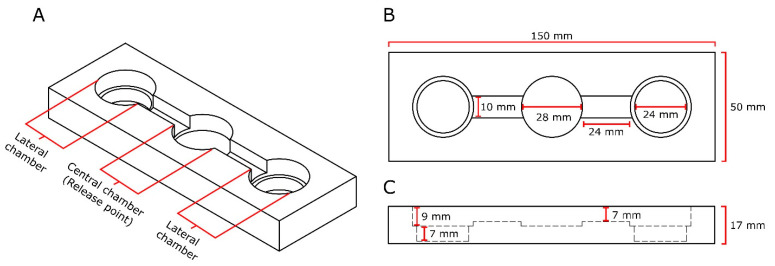
Graphical illustrations of lateral (**A**), overhead (**B**), and cross-section (**C**) views of the olfactometer used in tick bioassays. For bioassays, the lateral chambers of the olfactometer were filled with (*i*) treatment or control substrate (5 mL each; Exps. 1–25) or (*ii*) fitted with a watch glass holding a piece of filter paper treated with a test or a control stimulus (Exps. 26–31). Olfactometers were sealed with parafilm and a rectangular lid (150 × 50 × 3 mm). For each bioassay replicate, a single tick was released into the central chamber and was considered a responder if it was present in a lateral chamber at the end of the bioassay.

**Figure 2 microorganisms-12-00609-f002:**
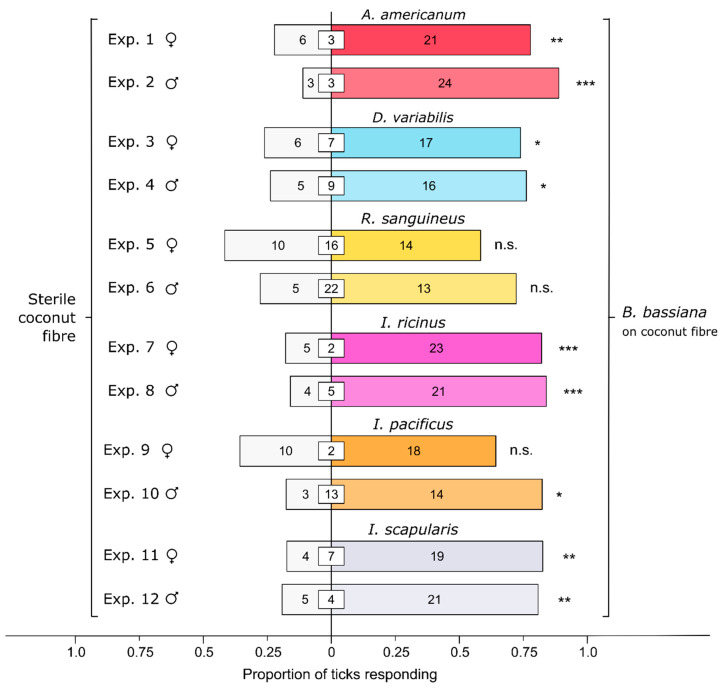
Effect of a *Beauveria bassiana* spore suspension on behavioral responses of ticks in six species: *Amblyomma americanum*, *Dermacentor variabilis*, *Rhipicephalus sanguineus*, *Ixodes ricinus*, *Ixodes pacificus*, and *Ixodes scapularis*. In olfactometer bioassays (Figure 1), ticks were offered a choice between coconut fiber treated or not treated (control) with a 1-mL *B. bassiana* spore suspension (10^7^ conidia/mL; 0.05% Tween80) incubated for 24 h. The control coconut fiber was treated with sterile 0.05% Tween80 (1 mL). The numbers in the bars represent the total number of ticks choosing a stimulus, and the numbers in the white inset boxes represent the total number of non-responding ticks. The asterisks indicate significant arrestment behavior on coconut fiber treated with a fungal spore suspension (exact binomial tests; * *p* < 0.05, ** *p* < 0.01, *** *p* < 0.001; n. s. = not significant).

**Figure 3 microorganisms-12-00609-f003:**
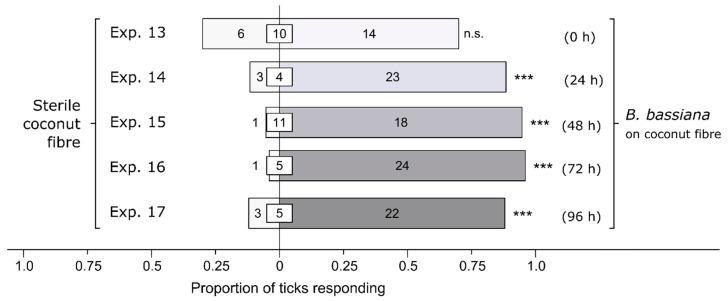
Effect of incubation time of a *Beauveria bassiana* spore suspension on behavioral responses of female black-legged ticks, *Ixodes scapularis.* In olfactometers bioassays (Figure 1), ticks were offered a choice between coconut fiber treated or not treated (control) with a 1-mL *B. bassiana* spore suspension (10^7^ conidia/mL; 0.05% Tween80) incubated for 0, 24, 48, 72, or 96 h before bioassays. The control coconut fiber was treated with sterile 0.05% Tween80 (1 mL). The numbers in the bars represent the total number of ticks choosing a stimulus, and the numbers in the white inset boxes represent the total number of non-responding ticks. The asterisks indicate significant arrestment behavior on coconut fiber treated with a fungal spore suspension (exact binomial tests; *** *p* < 0.001; n. s. = not significant).

**Figure 4 microorganisms-12-00609-f004:**
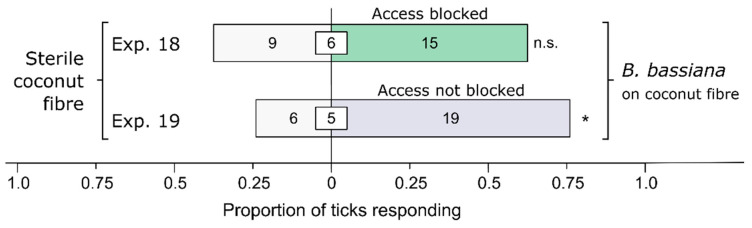
Effect of stimulus accessibility on behavioral responses of female black-legged ticks, *Ixodes scapularis.* In olfactometer bioassays (Figure 1), ticks were offered a choice between coconut fiber treated or not treated (control) with a 1-mL *Beauveria bassiana* spore suspension (10^7^ conidia/mL; 0.05% Tween80) incubated for 24 h before bioassays. The control coconut fiber was treated with sterile 0.05% Tween80 (1 mL). The stimuli in lateral chambers were accessible (Exp. 18) and inaccessible (Exp. 19), respectively. The numbers in the bars represent the total number of ticks choosing a stimulus, and the numbers in the white inset boxes represent the total number of non-responding ticks. The asterisk indicates significant arrestment behavior on accessible coconut fiber treated with a fungal spore suspension (exact binomial tests; * *p* < 0.05; n. s. = not significant).

**Figure 5 microorganisms-12-00609-f005:**
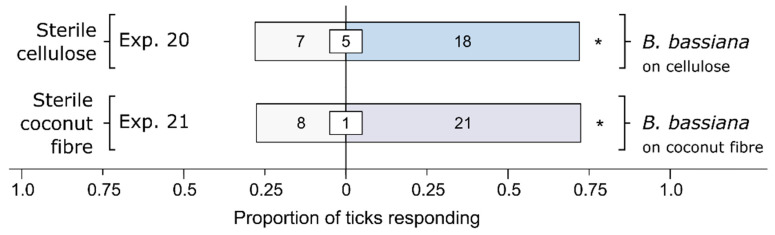
Effect of substrate (cellulose or coconut fiber), treated with a *Beauveria bassiana* spore suspension and incubated for 24 h, on behavioral responses of female black-legged ticks, *Ixodes scapularis*. In olfactometer bioassays (Figure 1), ticks were offered a choice between a substrate treated or not treated (control) with a 1-mL spore suspension of *B. bassiana* (10^7^ conidia/mL; 0.05% Tween80) incubated for 24 h. The control substrate was treated with sterile 0.05% Tween80 (1 mL). The numbers in the bars represent the total number of ticks choosing a stimulus, and the numbers in the white inset boxes represent the total number of non-responding ticks. The asterisks indicate significant arrestment behavior on the substrate treated with a fungal spore suspension (exact binomial tests; * *p* < 0.05; n. s. = not significant).

**Figure 6 microorganisms-12-00609-f006:**
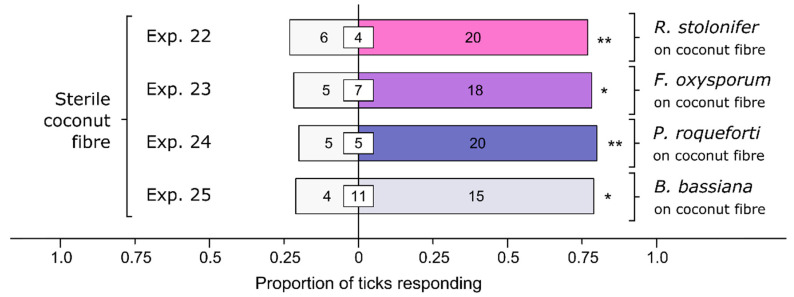
Effect of fungal species on arrestment behavior of female black-legged ticks, *Ixodes scapularis*. In olfactometer bioassays (Figure 1), ticks were offered coconut fiber treated or not treated (control) with a 1-mL spore suspension of *Rhizopus stolonifer*, *Fusarium oxysporum*, *Penicillium roqueforti*, or *Beauveria bassiana* (each 10^7^ conidia/mL; 0.05% Tween80) incubated for 24 h. The control coconut fiber was treated with sterile 0.05% Tween80 (1 mL). The numbers in the bars represent the total number of ticks choosing a stimulus, and the numbers in the white inset boxes represent the total number of non-responding ticks. The asterisks indicate significant arrestment behavior on coconut fiber treated with a fungal spore suspension (exact binomial tests; * *p* < 0.05, ** *p* < 0.01; n. s. = not significant).

**Figure 7 microorganisms-12-00609-f007:**
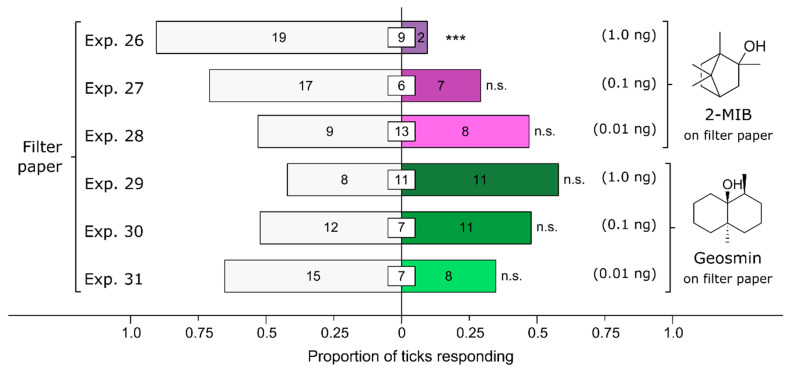
Effect of test chemicals applied on filter paper on behavioral responses of female black-legged ticks, *Ixodes scapularis.* In olfactometer bioassays (Figure 1), ticks were offered a choice between filter papers treated with (*i*) 2-methylisoborneol (2-MIB) dissolved in methanol and a methanol control, and (*ii*) geosmin in methanol and a methanol control. The numbers in the bars represent the total number of ticks choosing a stimulus, and the numbers in the white inset boxes represent the total number of non-responding ticks. The asterisks indicate significant avoidance of filter paper treated with 2-MIB at 1 ng (exact binomial tests; *** *p* < 0.001; n. s. = not significant).

## Data Availability

The data presented in this study are available through Mendeley Data at http://doi.org/10.17632/6k92byyrzp.1.

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
