# Peer review of "Harmful and Harmless Soil-Dwelling Fungi Indicate Microhabitat Suitability for Off-Host Ixodid Ticks"

_microorganisms, 2024, doi:10.3390/microorganisms12030609_

Round 1

Reviewer 1 Report

Comments and Suggestions for Authors

This manuscript describes a study that sought to test if blood fed ticks avoided moist microhabitats with entomopathogenic fungi Beauveria bassiana. They conducted experiments using six species of ixodid ticks and tested their responses in an olfactometer bioassays. They found that all ticks oriented towards the moist areas with Beauveria bassiana and other harmless fungi (did not discriminate harmful or not). They also found out that a common fungal metabolite found in disturbed soils strongly deterred ticks.  

Overall, the paper is well written, and the experiments well designed and executed. The findings have been explained well and in detail. Excellent work. I’d encourage the authors to conduct field work to see if the results would hold in the actual field environment.

Author Response

Thank you for your assessment of our manuscript. Please see the attachment for our responses to your comments.

Reviewer 2 Report

Comments and Suggestions for Authors

Review of the article “Harmful and harmless soil-dwelling fungi indicate microhabiat suitability for off-host ixodid ticks”.

Biologists recently discovered that soil actinobacteria repel predatory worms by releasing 2-methylisoborneol, as well as geosmin, a substance that is responsible for the characteristic smell of earth after rain.This is how the bacteria warn predators that they are poisonous.A substance called geosmin is responsible for the characteristic smell of damp earth.It is produced by many soil microorganisms, including actinobacteria, cyanobacteria and fungi.The role of geosmin in soil ecosystems is just beginning to be elucidated.Geosmin, as well as another terpene, 2-methylisoborneol, attract red fire ants, since the actinobacteria that secrete these substances suppress the growth of fungi that are dangerous to the ants.Scientists have established another role for geosmin in soil ecosystems.It turned out that actinobacteria use it to repel predators - in laboratory conditions these were the roundworms Caenorhabditis elegans.Biologists have found that the presence of geosmin and 2-methylisoborneol in Petri dishes changes the pattern of movement of coenorabditis, causing them to move faster and change direction more often.The researchers decided to determine how the worms perceive these substances.For this purpose, two types of mutants were used: with impaired sense of smell and with impaired taste perception.It turned out that the first mutants continued to respond to terpenes, but the second ones did not perceive them.So scientists found that taste neurons are responsible for the perception of these substances.Experiments have shown that the terpenes themselves are not toxic to worms, but the studied bacteria S. coelicolor also secrete toxic metabolites that are dangerous for coenorabditis.Thus, geosmin and 2-methylisoborneol signal that the bacteria are poisonous.Scientists conclude that these terpenes in the bacterial world act as an analogue of the warning coloration of poisonous animals.2-methylisoborneol is a common fungal metabolite, and its increased presence in soils can significantly repel ticks.This compound has attracted attention in the field of biological tick control because it has potential repellent properties.The synthesized 2-methylisoborneol may be of commercial interest as a tick repellent.It is important to note that ticks differ in their biological characteristics and preferences in choosing a host.Therefore, for 2-methylisoborneol to become a widely used repellent, it is necessary to ensure that its repellent properties cover the maximum number of tick taxa.Future studies will be required on the effects of synthetic 2-methylisoborneol on various species of ticks to establish its effectiveness and safety.In addition, the composition and concentration of the repellent may need to be optimized to achieve maximum effectiveness.Fortunately, there is currently significant interest in developing new and safe tick repellents, which may facilitate further research and development of synthetic 2-methylisoborneol as a potential commercial tick repellent. 

The article is very well written and the information is very interesting for scientists. But there are minor points that need to be completed. 

1) From row - 7 and everywhere in the text the name of Beauveria bassiana must be abbreviated according to the International Classification of Nomenclature - B. bassiana but not (Bb) and as it is first written in the text, you need to add the author and year - Beauveria bassiana (Bals.-Criv.) Vuill., 1912.

2) References - The list of references must be unified. Page numbers are missing or formatted differently. In some places the years are written in bold letters, in others they are not. 

Author Response

(The authors gave the same response as above.)

Reviewer 3 Report

Comments and Suggestions for Authors

The proposal of the manuscript is interesting, but it needs to undergo an extensive redesign. Without these adjustments, it is impossible to assess whether the results found are following the applied methodology and objectives. I have some comments about the introduction and materials and methods sections:

Introduction:

Why did the authors choose the tick species? What do they have in common?

There is a contradiction in some information regarding the habits and host search of ixodid ticks. Not all tested ticks are found in vegetation environments. Not necessary to cite about Argasidae ticks.

It is known that ticks in the environment search for pheromones for a place of ecdysis and survival. This context has not been addressed.

Which tick structures are responsible for chemoreception?

It is known that ticks are not social arthropods, try to bring examples of hematophagous arthropods, if possible

Is the objective to test spatial or non-spatial repellency?

Materials and methods:

The tick R. sanguineus recently underwent a taxonomic change. Do the authors prove that it is Rhipicephalus sanguineus and not Rhipicephalus linnaei?

Were the ticks kept in a climate-controlled chamber? Why were adults tested?

What is the purpose of using a still air olfactometer? To simulate which environment/behavior? Why still-air if the tick has anemotaxis? Would this be a new methodology? The authors do not mention references. Furthermore, to determine whether a compound was repellent or attractive, gold standard positive and negative controls for repellency are required. What was the distance between the central chamber and the lateral chamber? Why the 30-minute wait?

How do the authors guarantee that the response was olfactory or by contact? There is not enough information to understand the methodology applied to behavioral testing. The text should be rewritten.

Author Response

(The authors gave the same response as above.)
